# Role and effectiveness of telephone hotlines in outbreak response in Africa: A systematic review and meta-analysis

Noah T. Fongwen[1,2]*, Almighty Nchafack[1], Hana Rohan[3], Jason J. Ong[1,4], Joseph D. Tucker[1,5], Nadine Beckmann[3], Gwenda Hughes[3], Rosanna W. Peeling[1]

1 Faculty of Infectious and Tropical Diseases, Clinical Research Department, London School of Hygiene and Tropical Medicine, London, United Kingdom, 2 Division of Laboratory Systems and Networks, Africa Centres for Disease Control and Prevention, Addis Ababa, Ethiopia, 3 United Kingdom Public Health Rapid Support Team, London School of Hygiene and Tropical Medicine, London, United Kingdom, 4 Central Clinical School, Monash University, Melbourne, Australia, 5 University of North Carolina Institute of Global Health and Infectious Diseases, Chapel Hill, North Carolina, United States of America

* Noah.Fongwen@lshtm.ac.uk, FongwenN@africa-union.org

**Data Availability Statement:** All relevant data are within the paper and its supporting information files.

**Funding:** The author Noah T Fongwen was funded by the UK Public Health Rapid Support Team to

## Abstract

### Background

In Africa, little is known about the role of telephone hotlines in outbreak response. We systematically reviewed the role and effectiveness of hotlines on outbreak response in Africa.

### Method

We used the Cochrane handbook and searched five databases. The protocol was registered on PROSPERO (CRD42021247141). Medline, Embase, PsycINFO, Global Health and Web of Science were searched from 30 June 2020 to August 2020 for studies on the use of telephone hotlines in outbreak response in Africa published between January 1995 and August 2020. The search was also repeated on 16 September 2022. Data on effectiveness (alerts generated, cases confirmed) were extracted from peer-reviewed studies. Meta-analysis of alerts generated, and proportion of cases confirmed was done using the random effects model. The quality of studies was assessed using the Joanna Briggs Institute (JBI) tools. The heterogeneity and publication bias were assessed using the Galbraith and funnel plots, respectively.

### Results

Our search yielded 1251 non-duplicate citations that were assessed. 41 full texts were identified, and 21 studies were included in the narrative synthesis, while 12 were included in the meta-analysis. The hotlines were local (seven studies) or national (three studies). A combination of a local and national hotline was used in one study. The hotlines were set up for unusual respiratory events (one study), polio (one study), Ebola (10 studies), COVID-19 (two studies), malaria (one study), influenza-like illnesses (ILI) (one study) and rift valley fever in livestock (one study). Hotlines were mainly used for outbreak surveillance at the local level. A total of 332,323 alerts were generated, and 67,658 met the case definition,

carry out this study. The funders played no role in designing the study, collecting and analysing the data. Authors that represent the funding organisation were: Hana Rohan, Nadine Beckman and Gwenda Hughes, only reviewed versions of the manuscript.

**Competing interests:** The authors have declared that no competing interests exist.

corresponding to an overall pooled proportion of alerts generated(sensitivity) of 38% (95% CI: 24–52%). The sensitivity was 41% (95% CI: 24–59%) for local hotlines and 26%(95% CI:5–47%) for national hotlines. Hotlines were also used for surveillance of rift valley fever in livestock (one study) vaccination promotion (one study), death reporting (five studies), rumour tracking and fighting misinformation (two studies) and community engagement (five studies). The studies were of low to moderate quality with high publication bias and heterogeneity($I^2$ = 99%). The heterogeneity was not explained by the sample size.

## Conclusion

These data suggest that telephone hotlines can be effective in outbreak disease surveillance in Africa. Further implementation research is needed to scale up telephone hotlines in rural areas.

## Introduction

Mobile phones are simple but powerful devices that could enhance health services in Africa. Cell phone penetration in sub-Saharan Africa (SSA) has been steadily increasing [1]. At least 477 million people in SSA (45% of the population) subscribed to mobile phone services at the end of 2019 [2]. It is estimated that by the end of 2025, the number of smartphone connections in the region will almost double to reach 678 million as internet penetration and mobile phone ownership increase in the continent [3]. Public and private entities have used telephone hotlines to provide general health information and education and repurposed them as alert lines for outbreak response. For example, in Malawi, the Health Centre by Phone which was used to provide health information on nutrition and general health, was later repurposed for outbreak response during the COVID-19 pandemic [4,5]. There is no standard definition of a telephone hotline. However, in health care, a telephone hotline can be a toll-free call line or charged at a fixed cost to address the population's health needs.

Despite the potential of telephone hotlines, evidence of their role and impact during outbreaks is lacking. Most reviews of m-health have in developing countries have focused on the use of mobile phones in chronic disease management and health behavior changes in non-outbreak settings [6,7]. One literature review assessed African telephone hotlines for risk communication and community engagement in 13 African countries during outbreaks [8]. However, this study was not a systematic review and did not assess study quality. There was also a narrow focus on communication and COVID-19 in a few countries.

Data is needed on the effectiveness of telephone hotline during outbreaks in Africa. This evidence is of particular importance to the Africa Centres for Disease Control (Africa CDC), which was established to safeguard the health of the continent by detecting and responding quickly and effectively to disease threats and outbreaks [9]. Before Africa CDC or any organization can support telephone hotlines across Africa, evidence of the usefulness of hotlines is needed. This systematic review and meta-analysis assessed the published evidence on the role and effectiveness of telephone hotlines for outbreaks in Africa.

## Methods

### Protocol and registration

This review was conducted following the PRISMA (Preferred Reporting Items for Systematic Reviews and Meta-Analyses) statement of 2020 [10]. The protocol was registered in the

International Prospective Register of Systematic Reviews (PROSPERO). The registration number is CRD42021247141.

## Eligibility criteria

In this study, we included only studies in which telephone hotlines were set up primarily for outbreaks, repurposed from an existing toll-free line, or extended for use as part of an outbreak response. We defined a telephone hotline as a health hotline using a toll-free number that can enable the transmission of messages via calls and/or text (short messages) to prevent or respond to an outbreak of infectious diseases.

Studies were included if they: 1) were published between January 1995 and September 2022 where the last search was carried out; 2) mentioned a telephone hotline set up primarily for an outbreak response or was initially used for other purposes but either extended for use in an outbreak or repurposed entirely for an outbreak, and mentioned how the hotline was used with data on specified measures of effect such as the proportion of alerts generated, proportion of alerts generated that were tracked and finally tested positive, or the number of deaths reported; 3) reported qualitative effects such as communication, rumour tracking, dispelling of myths and combatting misinformation; 3) conducted in Africa; and 4) contained primary data. Reviews were excluded. There was no restriction on the language.

## Source of information and search strategy

Using medical subject headings (MeSH) and text keywords, the literature was searched relating to the use of telephone hotlines in outbreaks in Africa. The search was carried out on 30 June 2021 and updated on 16 September 2022. The search strategy was developed by the research lead (NF) and refined with the help of London School of Hygiene and Tropical Medicine librarians. Two independent individuals (NF and CA) searched citations in MEDLINE, Embase, PsycINFO, Global Health and Web of Science. The search strategy is in S1 File.

Google and Google scholar were searched for studies from each of the 54 African countries using the search term: ('hotline' or 'call centres' or 'mobile phone' OR 'toll-free number') and (outbreak* or epidemics) and (name of country).

The search outputs were exported to Mendeley desktop V.1.16.1, and duplicates were removed. After removing duplicates in Mendeley, the titles and abstracts of the studies were independently screened by two individuals. The full texts were obtained from the screened abstracts after inclusion and exclusion criteria were applied.

The reference lists of relevant full texts were searched to identify relevant studies. After identifying the relevant articles, the authors were contacted for any information on other similar studies published on telephone hotlines in Africa. Authors were also contacted in case of discrepancies related to studies published in the same country and about the same telephone hotline.

## Data collection and data items

A data extraction spreadsheet was developed in Excel version 2013. Data extracted on the excel sheet were on the general characteristics of the hotlines, the effect and cost. The general characteristics included the country, history of other hotlines, coverage of the hotline, the role of the hotline, use of data capture tools, and the study design. The measures of effect extracted were: i) the proportion of alerts generated by the hotline that met the case definition (defined as the number of the alerts generated using the hotline divided by the total number of alerts generated); ii) the proportion of confirmed cases identified through follow up with the hotline (defined as the number of cases confirmed through follow up with the hotline divided by the

total number of cases confirmed; iii) other outcomes reported were the proportion of deaths reported and vaccination uptake(encouraging and confirming vaccination). The 95% confidence intervals of these measures of effect were also calculated using generic formulas from the University of California San Francisco Clinical and Translational Science Institute.

## Analysis and synthesis of results

A narrative synthesis and a meta-analysis were done. In the narrative synthesis, the findings were critically analysed using the reach of the hotlines and their effectiveness as themes, taking into consideration the key methodological flaws in the studies. Meta-analysis was performed using the statistical software STATA V.17.0. The random effects model was used to pool the measures of effect from the studies. Two forest plots were produced for the proportion of alerts generated and the proportion of confirmed cases. These two forest plots each considered two sub-categories: national hotline vs local hotline. The heterogeneity in the effect sizes of the studies was assessed using $I^2$ values. A meta-regression was carried out to explain the heterogeneity. A funnel plot and the egger test were used to assess for publication bias. Only studies with complete data on the generation of alerts and/or cases were included in the meta-analysis. The studies with only data on death reporting or other outcomes, such as immunization rates, were not included in the meta-analysis. These studies were discussed separately.

## Quality assessment of studies

The quality of the studies was assessed using the Joana Briggs Institute (JBI) tools [11]. Cross-sectional studies were classified as low quality (score $\leq$4), moderate quality(score = 5–6) and high quality(score = 7–8). Cohort studies with scores $\leq$6 were of low quality.

# Results

## Search results

Fig 1 is a PRISMA flow diagram of our search. Our search yielded 1251 non-duplicate hits that were later screened using our inclusion and exclusion criteria. After screening, 41 full texts were retrieved. Finally, 21 studies were included in the narrative synthesis, while 12 were included in the meta-analysis.

## General characteristics of included studies

Table 1 is a summary of the included studies. Of the included studies, two were from southern Africa [12,13], four were from central Africa [14–17], six were from east Africa [18–23] and nine were from west Africa [24–32]. We found no studies from northern Africa. In terms of coverage, six were local and six were national hotlines. The hotlines were set up for unusual respiratory events (one study), polio (one study), Ebola (10 studies), COVID-19 (two studies), malaria (one study), influenza-like illnesses (ILI) (one study) and rift valley fever in livestock (one study). The hotlines in Sierra Leone and South Sudan were set up for Ebola initially but extended for use in the COVID-19 response. The hotlines were mainly used as part of the event-based surveillance systems in the countries. In most studies (12 out of 19), data were collected and reported using paper-based systems. Seven studies used digital reporting. In terms of study design, eight studies were prospective cohorts, ten were cross-sectional, two were mixed method studies, and one was a before/after study. Two studies explored the use of telephone hotlines in rumour tracking, raising awareness, dispelling myths, and combatting misinformation during outbreaks [20,27].

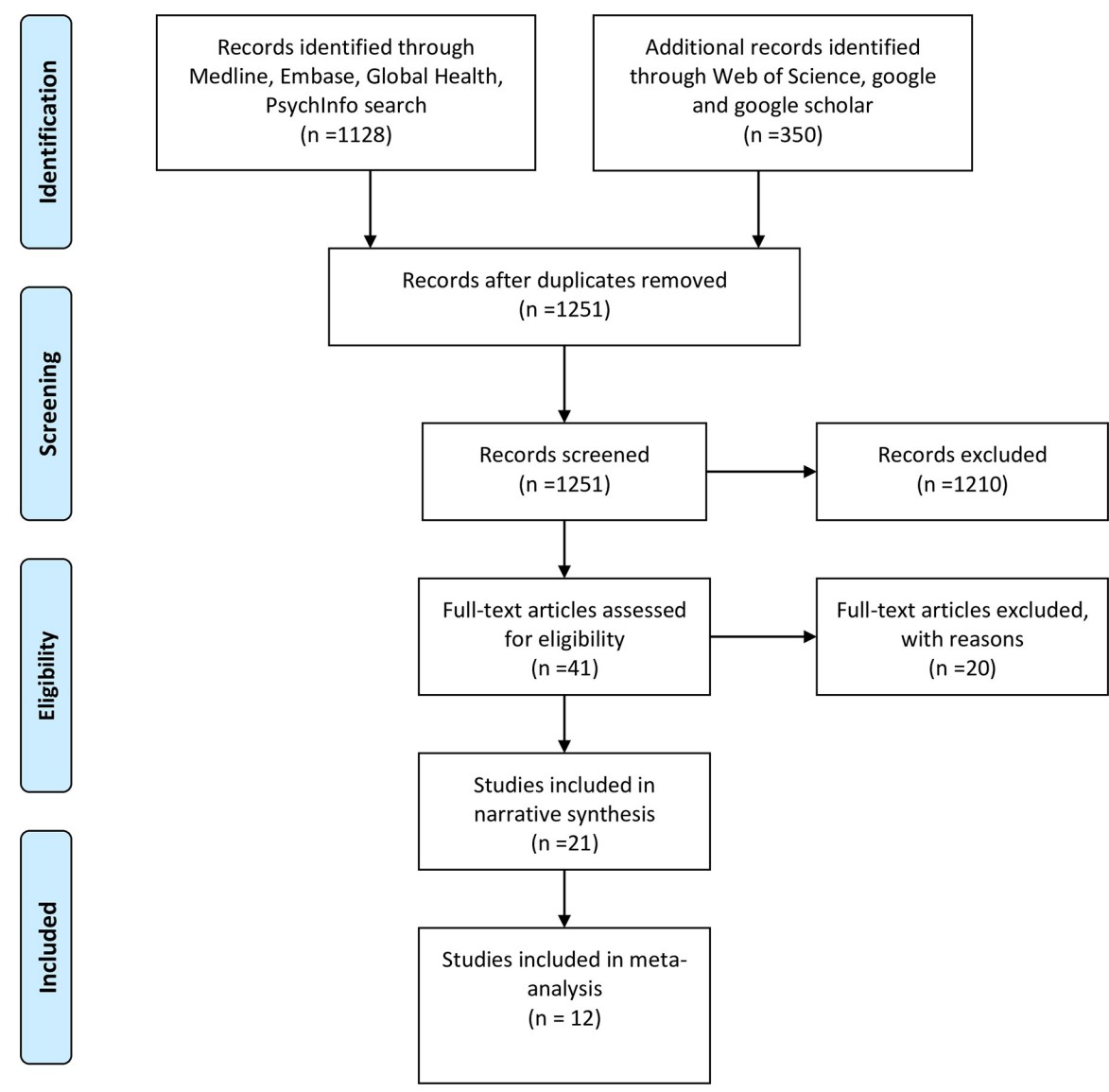

**Fig 1. PRISMA flow diagram showing the selection of studies.**

### Effectiveness of the hotlines

Table 2 shows the sensitivity of the hotlines in generating alerts. Fig 2 is a forest plot from the meta-analysis of the studies in Table 2. A total of 332,323 alerts were generated, and 67,658 met the case definition. These alerts corresponded to an overall pooled sensitivity of 38% (95% CI: 24–52%) $I^2$ = 99.99%. When stratified according to the coverage (local vs. national), the sensitivity was 41% (95% CI: 24–59%) $I^2$ = 99.94% for local hotlines and 26%(95%CI:5–47%) $I^2$ = 99.98% for national hotlines.

Table 3 shows the proportion of confirmed cases identified through follow-up with the hotline. Fig 3 is a forest plot from the meta-analysis of the studies in Table 3. Of the 2958 confirmed cases, 562 were identified through follow-up with the hotline. This corresponded to an overall pooled proportion of 47% (95%CI: 20–74%) $I^2$ = 99.53%. The meta-regression showed the heterogeneity was not explained by the sample size (S2 File).

**Table 1. Characteristics of included studies.**

| | Study | Country | Coverage of hotline | Origin, evolution, and use of hotline | Cost of the hotline | Data capture tool | Study design |
|---|---|---|---|---|---|---|---|
| 1 | Alroy et al, 2020 [14] | Cameroon | Local | The hotline was set up to report unusual respiratory events. No information on the use for any national outbreak. | Not stated | Paper based | Description of a prospective cohort study |
| 2 | Aluma et al, 2021 [15] | Chad | Local | The hotline was set up to increase vaccination coverage for polio. Plans for national scale up were stated but no information on its use for covid 19 was available. | Not stated | Paper based | Before after study |
| 3 | Keita et al, 2021 [17] | DR Congo | Local | The hotline was set up as part of the early warning and response system for Ebola. No information on its use for covid. | The minimum average cost of the system was US $438/ case detected and US $1.8/ alert received. | Paper based | Cross sectional |
| 4 | Hemingway-Foday et al, 2020 [16] | DR Congo | Local | The hotline was set up as part of a system to reinforce epidemic surveillance for Ebola virus in 2017. | Not stated | Paper based | Cross sectional |
| 5 | Coulibaly et al, 2019 [25] | Ivory Coast | Local | The hotline was used in Ebola event-based surveillance. | Not stated | Paper based | Cross sectional |
| 6 | Tibbels et al, 2021 [27] | Ivory Coast | National | For real time tracking of covid 19 rumors. | Not stated | Electronic | Descriptive |
| 7 | Oladeji et al, 2020 [22] | Ethiopia | Local | The hotline was set up for covid response and was used as an alert and surveillance system. | Not stated | MS excel spreadsheet | Cross sectional |
| 8 | Lee et al, 2016* [30] | Guinea | National and local | The hotline was set up for Ebola response. No information if it was repurposed for use in covid. | Not stated | Electronic record | Cross sectional |
| 9 | Oyas et al, 2018 [23] | Kenya | National | Telephone was set up for surveillance of rift valley fever in livestock. | Not stated | Not stated. Developed by the team | Prospective cohort study |
| 10 | Kouadio et al, 2015 [26] | Liberia | Local | The hotline was used by the Ebola emergency response for communities to report alerts. | Not stated | Electronic record | Prospective cohort study |
| 11 | Rajatonirina et al, 2012 [12] | Madagascar | National | The hotline used for sentinel surveillance for influenza-like illness (ILI) | Not stated | Electronic | Prospective cohort study |
| 12 | Karim et al, 2021 [20] | Rwanda | National | The hotline was used in rumour tracking and management, community engagement and awareness during the covid-19 pandemic. | Not stated | Electronic | Descriptive |
| 13 | Miller et al, 2014 [31] | Sierra Leone | National | The 117 is national hotline first used for Ebola (alert, surveillance, and death reporting). Decentralized to local regions. It was extended for use in covid-19. The hotline was used for alerts and surveillance for Ebola. | Not stated | Paper based | Prospective cohort study |
| 14 | Jia et al, 2015 [32] | Sierra Leone | Local | | Not stated | MS excel bulletin sheet | Prospective cohort study |
| 15 | Gashu et al, 2017 [28] | Sierra Leone | Local | | Not stated | Paper based | Prospective study |
| 16 | Alpren et al,2017 [24] | Sierra Leone | National | | With 198 staff at the peak, it cost $200,000 per month to run the hotline, which reduced to $47,000 as the pandemic waned. | Paper based | Cross sectional |
| 17 | Jalloh et al, 2020 [29] | Sierra Leone | National | | Not stated | Paper based | Cross sectional |
| 18 | Davies et al, 2019 [13] | South Africa | National | This hotline system was set up for malaria alerts and surveillance. | Not stated | MalariaConnect system and paper based | Prospective cohort |

**Table 1.** (Continued)

| | Study | Country | Coverage of hotline | Origin, evolution, and use of hotline | Cost of the hotline | Data capture tool | Study design |
|---|---|---|---|---|---|---|---|
| 19 | Olu et al, 2019 [18] | South Sudan | National | The '6666' was set up for ebola response and was later repurposed for use in covid. | Not stated | Paper based | Mixed method |
| 20 | Lopez et al, 2021 [19] | South Sudan | National | | Not stated | Paper based | Cross sectional |
| 21 | Kok et al, 2021 [21] | Uganda | (Local with focus on rural communities) | The hotline was used within a call centre for community health workers to report cases of covid 19 and for community engagement. | Not stated | Paper based | Mixed method |

Fig 4 is a funnel plot to assess for publication bias. From the figure, the points are more concentrated in larger effect sizes. The publication bias was high toward positive effects. The egger test also confirms the high publication bias (S3 File).

Other measures of effect reported were the rate of death reporting (from three studies) and vaccination confirmation (from one study). From the studies that used hotlines in reporting deaths and for mortality surveillance, the rate of reporting via the hotline ranged from 34% to 83%, depending on the stage of the outbreak. The only study that used a telephone hotline to follow up on vaccination coverage was conducted in Chad. This study showed that by using telephone calls to verify and confirm vaccinators' visits, the vaccination coverage increased from 43% before intervention to 95% after the intervention. This corresponded to a significant attributable change of 0.52% (95% CI: 0.51–0.53) in the vaccination coverage for polio in children.

One study from Kenya used telephone hotlines in surveillance for rift valley fever (RVF) in livestock during El Nino rains and the threat of RVF. In this study, 69 herds met the case definition for RVF, and 24 were probable cases. There was no follow-up confirmation of cases.

In two studies, the effect reported was in terms of the number of rumors submitted. One of the studies, conducted in Ivory Coast, used the telephone hotline to submit 1,747 rumors

**Table 2. Effectiveness of hotlines: Proportion of alerts that met case definition.**

| Author | Country | Number of live alerts | Number of alerts meeting case definition through hotline | Proportion of alerts generated by hotline (95% CI) |
|---|---|---|---|---|
| Keita et al, 2021 [17] | DR Congo(local) | 194,798 | 30,738 | 0.1578(0.1562–0.1594) |
| Oladeji et al, 2020 [22] | Ethiopia(local) | 259 | 128 | 0.4942(0.4318–0.5568) |
| Lee et al, 2016 [30] | Guinea (National) | 17309 | 1778 | 0.1027(0.0982–0.1073) |
| Lee et al, 2016 [30] | Guinea(local) | 8667 | 5006 | 0.5776(0.5671–0.5880) |
| Jia et al, 2015 [32] | Sierra Leone(local) | 260 | 129 | 0.4962(0.4338–0.5586) |
| Gashu et al, 2017 [28] | Sierra Leone(local) | 11303 | 7059 | 0.6245(0.6155–0.6335) |
| Miller et al, 2014 [31] | Sierra Leone (National) | 3299 | 1202 | 0.3644(0.3479–0.3810) |
| Coulibaly et al, 2019 [25] | Ivory Coast (Local) | 92 | 10 | 0.1087(0.0534–0.1908) |
| Kouadio et al, 2015 [26] | Liberia(local) | 737 | 619 | 0.8399(0.8114–0.8656) |
| Rajatonirina et al, 2012 [12] | Madagascar (National) | 95401 | 20933 | 0.2194(0.2168–0.2221) |
| Kok et al, 2021 [21] | Uganda(local) | 91 | 5 | 0.0549(0.0181–0.1236) |
| Olu et al 2019 [18] | South Sudan (national) | 107 | 51 | 0.4766(0.3792–0.5754) |
| Total | | 332,323 | 67,658 | |

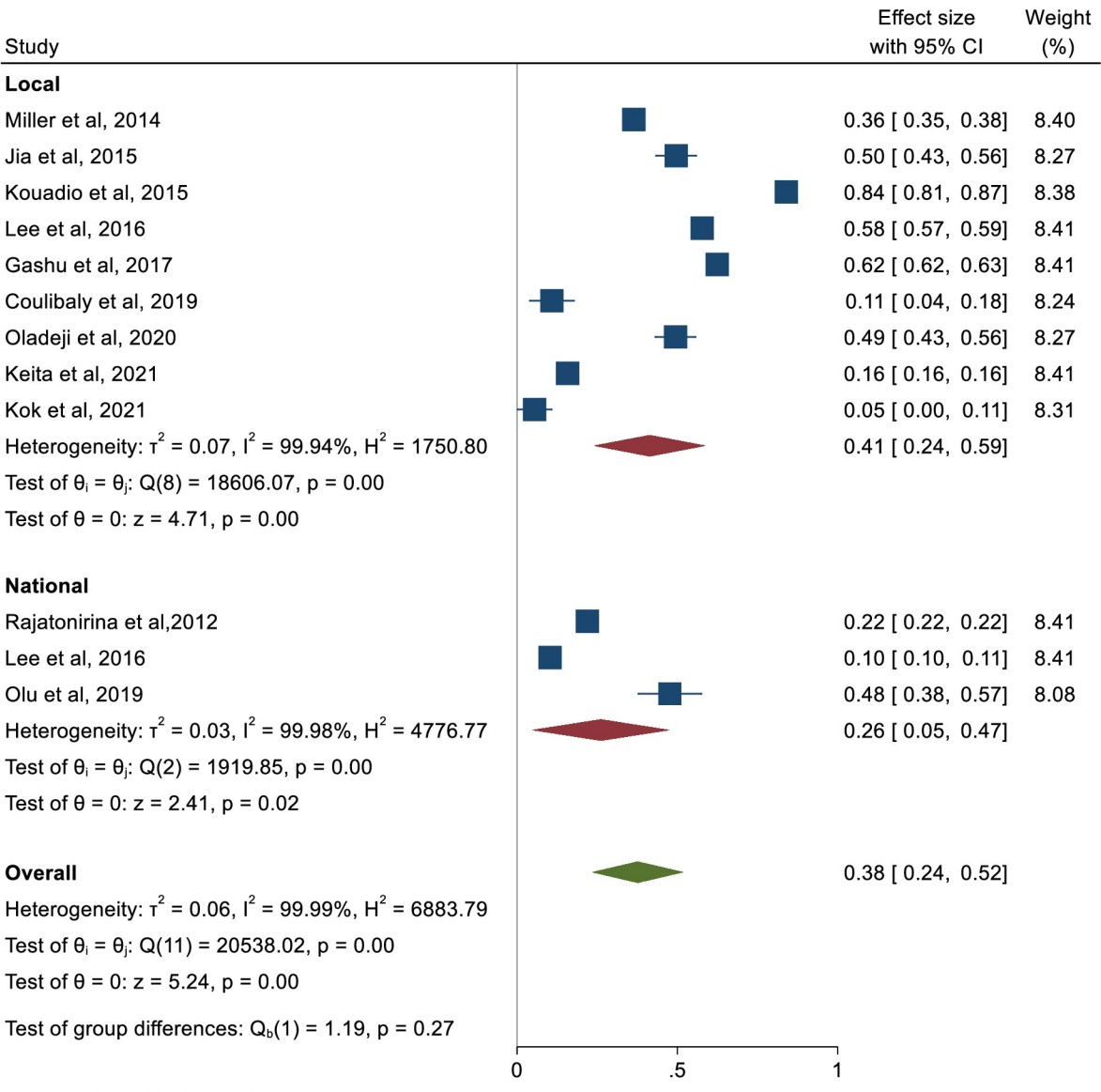

**Fig 2. Forest plot showing pooled sensitivity in generating alerts by coverage of the hotline.**

**Table 3. Effectiveness of hotline: Proportion of confirmed cases generated using the hotline.**

| Author | Country | Number of confirmed cases from hotline alerts | Total number of confirmed cases | Sensitivity in case detection(95% CI) |
|---|---|---|---|---|
| Keita et al, 2021 [17] | DR Congo(local) | 300 | 801 | 0.3745(0.3409–0.4091) |
| Oladeji et al, 2020 [22] | Ethiopia(local) | 22 | 43 | 0.5116(0.3546–0.6669) |
| Lee et al, 2016 [30] | Guinea(National) | 71 | 1,838 | 0.0386(0.0303–0.0485) |
| Lee et al, 2016 [30] | Guinea(local) | 120 | 221 | 0.5430(0.4749–0.6100) |
| Jia et al, 2015 [32] | Sierra Leone (local) | 49 | 55 | 0.8909(0.7775–0.9589) |
| | | 562 | 2958 | |

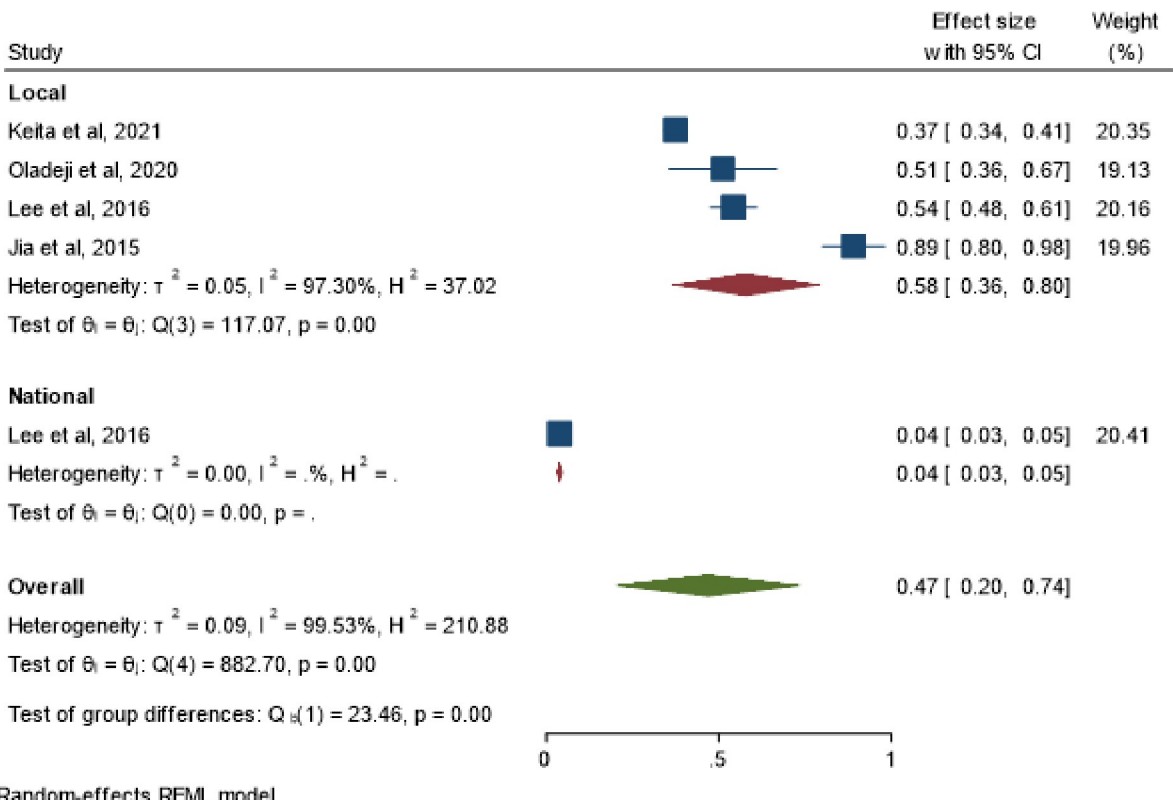

**Fig 3. Forest plot showing the pooled proportion of cases identified through follow up with the hotline.**

coded, analyzed, and used to inform risk communication and community engagement activities.

## The quality of included studies

S4 File and S5 File show the results of the quality assessment of the studies. The cross-sectional studies were of low to moderate quality. The cohort studies were of low quality.

## Cost of the hotlines

Only two studies analyzed the cost of operating the hotlines. One study in the DR Congo presented the cost per case detected and the cost per alert received. From this study, it cost the system a minimum of $438 to detect one case and $1.8 to receive one alert over a 2-year period. There was no mention of the number of staff or the phase of the outbreak in which these estimates were made. In another study in Sierra Leone, the average cost of running the system per month was reported to be $200,000 per month with a maximum of 198 staff employed at the peak of the outbreak. The cost reduced to $47,000 during the off-peak phase. However, none of these studies clearly outlined the set-up costs and maintenance costs of the hotlines.

## Discussion

Telephone hotlines and other emerging digital technologies have been identified as an essential area of epidemic response [33]. Our study delved into the critical role played by telephone

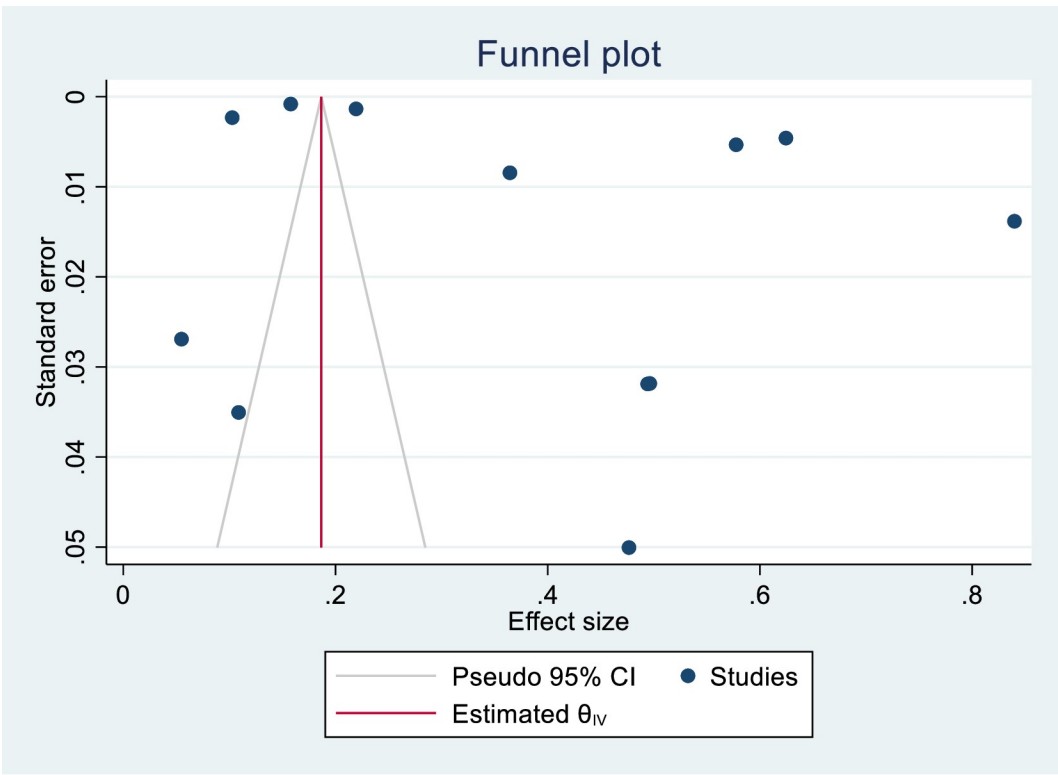

**Fig 4. Funnel plot for publication bias.**

hotlines and the assessment of the effect. Our findings demonstrate that many African countries have used hotlines to enhance disease surveillance. This study extends the literature by examining hotlines in LMIC settings, using a systematic review, and meta-analysis of data from 12 studies.

Our study shows that telephone hotlines can play a crucial role in disease surveillance in the generation of alerts during outbreaks by providing a quick and easy means for individuals to report potential outbreaks and seek information and guidance. In the early stages of an outbreak when timely detection and response can limit the spread of disease, this ability of rapidly generating alerts is extremely useful. Other studies have been conducted outside Africa to examine the effectiveness of telephone hotlines in generating alerts and predicting outbreaks. In a large metropolitan area in Japan, Katayama et al showed that the hotline could be effective in predicting an outbreak of influenza [34]. They showed that there was a high level of correlation between the telephone triages and the number of influenza patients, making it possible to use the hotline for early alert generation and response. Other studies in the United Kingdom have shown that telephone hotlines as part an early warning system successfully detected influenza-like illness outbreaks and abnormal rises in vomiting and diarrhoea, prompting a timely response from the public health authorities [35]. However, the timeliness of the response depends on the availability of resources. In one study from a resource constrained country like Yemen, there was instead a delay of about 3 days in the response to cholera when the national telephone hotline system was used [36].

Even though the overall evidence suggest that telephone hotlines can be effective in generating alerts, it is worth noting that some studies have provided evidence to refute this finding. In the United Kingdom, one study found that the national health service hotline can overlook

small, localized outbreaks and generate false positive alerts leading to the generation of unnecessary responses [37]. In addition, for diseases such as Cryptosporidiosis and Rotavirus, the hotline failed to detect outbreaks and resulted in bias reporting [37,38].

Overall, our study shows that while national hotlines can be effective at providing general information, local hotlines can be more effective at generating alerts. Taking into consideration specific disease outbreaks, resources available and the needs of the local community, local hotlines can be more effective at connecting individuals to local resources and facilitating communication between the public and local health departments. In Africa, at least 45 member states have national hotlines that receive calls and provide essential information to keep the public safe. In South Korea, the national hotline was used to provide reliable and accurate information about COVID-19 to the public [39]. These national hotlines easily become overwhelmed and lack adequate resources for follow up actions. The decentralization of national hotlines to local hotlines provides unique opportunities to get closer to the local communities, track cases, provide real-time alerts and easily adapted to implement effective disease control measures during the Ebola outbreaks Sierra Leone and South Sudan [18,28,31,32,39]. In China, during the COVID-19 pandemic, out of the 625 hotlines that were created across the country, 420 were rapidly adapted to meet the increasing the psychological demands [40].

Another important finding from our study is the relevance of telephone hotlines in reporting deaths. Telephone hotlines can be useful in mortality surveillance, particularly in low-resource settings where traditional surveillance systems may be inadequate. In many of these low-income countries, many deaths in remote areas can go unreported. The lack of complete and robust mortality data limits the ability of many countries in Africa to measure the impact of programmatic interventions and adequately plan to address their health needs [41]. While initiatives like the Rapid Mortality Mobile Phone Surveys(RaMMPS) consortium is developing innovative mobile phone approaches to generate national mortality estimates(including excess mortality) during outbreaks [42], we found that there is still a limited number of published literature on the role of telephone hotlines in mortality surveillance during outbreaks in Africa. All the studies included in our systematic review with data on deaths reported were from the Sierra Leone hotline. Out of Africa, researchers in Malaysia showed that it was feasible to use mobile phones to support mortality surveillance during the COVID-19 pandemic where face-to-face contacts were severely restricted [43]. Also, in non-outbreak settings, some studies have shown the usefulness of mobile telephones. One study in Papua New Guinea showed that health workers successfully used mobile phones for notification of deaths and verbal autopsy as part of their mortality surveillance programme adopted by the government [44]. In Malawi, another study found that more than half the deaths reported using mobile phones were not registered in the National Registration Bureau [45].

Despite the immense potential of mobile telephones in strengthening mortality surveillance in communities, their success will depend on a wide range of factors such as the availability of committed community health workers for routine reporting; the presence of established death management systems that involve contact with families; and the presence of reliable platforms that can support remote connectivity [46]. These factors also determine the quality and consistency of the data reported.

Regarding risk communication during pandemics, our study shows that by providing a direct line of communication between the public and trusted sources of information, telephone hotlines can be used to track rumours, dispel myths, and fight misinformation. Using telephones, individuals can report rumours that can be used to optimize effective public health messaging. Although countries like Comoros, Kenya, Madagascar, Malawi and Zambia used their telephone hotlines to track rumours during the COVID-19 pandemic [47], we found only one peer-reviewed study in Ivory Coast which systematically showed how rumours could

be tracked using telephone hotlines [27]. Conventional ways of tracking rumours have been through social media applications like twitter, Facebook, and Instagram. The World Health Organization (WHO) has established a rumour tracking programme and a community engagement hub. These WHO hubs use the leading social media outlets in rumour tracking. However, not all mobile phone users in developing countries possess telephones that can accommodate social media apps, and many older phone users may not be actively engaged on social media [48]. Consequently, telephone hotlines provide a simple means to extend the reach for rumour tracking to involve wider demography of users and those who may not have smartphones.

In all the studies that were included in this systematic review, the objective and rigorous measurement of the effectiveness of the hotlines posed a significant challenge. The need for appropriate tools and the capacity of the hotline operators to conduct basic research that assesses the effect of hotlines were essential barriers. No tool was available for the objective assessment of the effect of the hotlines in rumour tracking, fighting misinformation, and promoting community engagement. For studies that measured the effect of the hotlines in terms of the alerts generated and alerts that were subsequently followed up and tested, significant methodological flaws were identified. The cross-sectional studies used secondary data to assess the effect and did not adjust for confounding factors such as the age distribution, location of the callers, differences in socioeconomic status and health systems. In addition, prospective cohort studies had no initial considerations to ensure that the design and data collection methods were robust, and the comparator groups were unclear in most studies. In these cohort studies, there was no consideration of the effect of low response rates which resulted from callers not calling back or the hotline operators not doing appropriate follow-up. Most studies needed to outline the selection of sites and sampling frame clearly. These limitations may arise because there is usually more emphasis on implementing the intervention than basic research to generate evidence that may optimize the intervention in the setting of outbreaks [49]. It is, therefore, necessary to develop simple research tools and frameworks that can be used by hotline operators who are not usually researchers, to help them design, collect, analyze, and use the data collected while the outbreak is ongoing. These data will help them understand how well they are performing and help funders adequately direct funding to appropriate components of the hotline.

Our study shows that many call centres still depend on paper-based tools to collect, analyze, and report data. However, some call centres have used electronic data capture and software to ensure they handle data more efficiently. Paper-based tools may lead to delays in collection, analysis, and interpretation. In one study from Ivory Coast, the authors demonstrated that software such as the District Health Information System 2 (DHIS2) could collect and analyze rumours. The DHIS2 is widely available in many health districts across Africa [50] and could be considered a potential open-access software that can be used in a network of African hotlines by the Africa CDC to standardize the collection, analysis, and visualization of data for a more coordinated outbreak response. Furthermore, when using telephone hotlines, a considerable amount of data can be generated that may require more innovative and more efficient ways of analysis and interpretation. Machine learning and artificial intelligence could increase efficiency by making sense of the data collected by the hotlines. None of the studies mentioned the use of artificial intelligence, but in countries such as Togo, machine learning and phone data have been used to improve the targeting of humanitarian aid [51].

In general, randomized controlled trials (RCTs) will not be a suitable study design to show the effectiveness of telephone hotlines because they are not practical for such population-wide interventions and not relevant for outbreaks since decisions need to be made quickly [52]. In peace times, though expensive, a pragmatic stepped-wedged RCT can be used with a new

network expansion city by city. A before-and-after study design can be used to assess the effectiveness and impact of the hotlines. In Chad, Aluma *et al.* used a before-and-after study design to show that a polio hotline can increase polio vaccination uptake [15]. Even though there was no adjustment for confounders, the study shows that with training and attention to basic research, it is possible for research to generate quality evidence in call centres.

## Implications for policy and practice

Africa CDC Public Health Research pillar has as a key target to conduct operational research and evaluate the utility of mobile technologies as potential surveillance and information dissemination tools [53]. Our findings can help them understand the role of telephone hotlines and their effect on outbreak response. Lessons from the study designs and their limitations will be essential in guiding Africa CDC to better design and pilot future studies that can generate more robust evidence on the effectiveness of telephone hotlines during outbreaks and in peaceful times. In addition, the Africa CDC mortality surveillance program can benefit from the role of hotlines in reporting deaths during outbreaks. We found that in some countries, hotlines also played a role in post-outbreak mortality surveillance as they integrated into the national surveillance programs. Furthermore, our study shows that telephone hotlines can be used in outbreak response in livestock, which provides new opportunities to strengthen the Africa CDC One health strategies. Finally, telephone hotlines can be pivotal in communication and rumour tracking. Africa CDC can use a network of telephone hotlines in Africa to prevent the spread of 'infodemics' and control outbreaks.

## Strengths and limitations of the study

This systematic review is one of the few on mHealth strategies in Africa and the first on the role of telephone hotlines in outbreak response. Our methodology was rigorous as it involved independent search and screening of articles, use of multiple databases, hand searching, and contacting of authors. Despite the evidence of a moderate effect on alert generation from meta-analysis, it should be noted that the studies included in the meta-analysis needed to be better designed to generate robust evidence, and the publication bias was high towards positive effects. Therefore, the findings of the meta-analysis cannot be generalized. Despite this limitation, our study represents a first attempt to pool together the evidence on the effect of hotlines. It will encourage hotlines to develop a culture of generating programmatic evidence on the effect that can be used for decision-making. Furthermore, only quantitative measures of effect were considered. Only two studies describe the qualitative effects of hotlines in rumor tracking, dispelling myths, and fighting misinformation in the setting of outbreaks [20,27].

## Recommendations for future research

More studies using robust and pragmatic designs, such as quasi-experimental and pragmatic randomized controlled trials, should be used in generating evidence on the effectiveness of telephone hotlines. These effectiveness studies will provide a stratification of the different components of the hotline intervention, which will help refine the operations to optimize productivity and effectiveness. Qualitative studies on the effectiveness of hotlines need to be conducted to provide a holistic picture of their performance in the setting of outbreaks. More costing studies will be needed to better understand the financial implications of running and sustaining telephone hotlines in outbreak settings.

## Conclusion

Our study suggests that telephone hotlines can be effective in outbreak disease surveillance in Africa. Further implementation research with robust quantitative and qualitative methods is needed to scale up telephone hotlines in rural areas.

## Supporting information

**S1 Checklist. PRISMA 2020 checklist.**
(DOCX)

**S1 File. Database search strategy.**
(DOC)

**S2 File. Results of metaregression using sample size.**
(DOCX)

**S3 File. Results of egger test for publication bias.**
(DOCX)

**S4 File. Assessment of the quality of cross-sectional studies.**
(DOCX)

**S5 File. Assessment of the quality of cohort studies.**
(DOCX)

## Acknowledgments

The authors thank the London School of Hygiene and Tropical Medicine library for access the OVID search databases. The authors also thank the Dr Nafiisah Chotun and Dr Mohammed Abdullaziz of the division of disease control of Africa CDC, for their support and guidance during this project. Special thanks to Dr Hana Rohan for her guidance during this project.

## Author Contributions

**Conceptualization:** Noah T. Fongwen, Rosanna W. Peeling.

**Data curation:** Noah T. Fongwen.

**Formal analysis:** Noah T. Fongwen.

**Funding acquisition:** Noah T. Fongwen.

**Investigation:** Noah T. Fongwen.

**Methodology:** Noah T. Fongwen, Almighty Nchafack, Jason J. Ong, Joseph D. Tucker, Rosanna W. Peeling.

**Project administration:** Noah T. Fongwen.

**Resources:** Noah T. Fongwen.

**Software:** Noah T. Fongwen.

**Supervision:** Noah T. Fongwen, Jason J. Ong, Joseph D. Tucker, Nadine Beckmann, Gwenda Hughes, Rosanna W. Peeling.

**Validation:** Noah T. Fongwen.

**Visualization:** Noah T. Fongwen.

**Writing – original draft:** Noah T. Fongwen.

**Writing – review & editing:** Noah T. Fongwen, Hana Rohan, Jason J. Ong, Joseph D. Tucker, Nadine Beckmann, Gwenda Hughes, Rosanna W. Peeling.

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
