## [Decision Letter · Decision Letter 0]

24 Aug 2023

PONE-D-23-02720Effectiveness of telephone hotlines in outbreak response in Africa: a systematic review and meta-analysisPLOS ONE

Dear Dr. Fongwen,

Thank you for submitting your manuscript to PLOS ONE. After careful consideration, we feel that it has merit but does not fully meet PLOS ONE’s publication criteria as it currently stands. Therefore, we invite you to submit a revised version of the manuscript that addresses the points raised during the review process.

We look forward to receiving your revised manuscript.

Kind regards,

Felix Bongomin, MB ChB, MSc, MMed, FECMM

Academic Editor

PLOS ONE

Journal Requirements:

Reviewers' comments:

Reviewer's Responses to Questions

**Comments to the Author**

1. Is the manuscript technically sound, and do the data support the conclusions?

Reviewer #1: Yes

2. Has the statistical analysis been performed appropriately and rigorously? 

Reviewer #1: Yes

3. Have the authors made all data underlying the findings in their manuscript fully available?

Reviewer #1: Yes

4. Is the manuscript presented in an intelligible fashion and written in standard English?

Reviewer #1: Yes

5. Review Comments to the Author

Reviewer #1: Thank you for the opportunity to review this paper

The paper is quite well written, below are my comments.

Comments

Abstract

Background:

1. The authors state “In Africa, little is known about the role of telephone hotlines in outbreak response. We systematically reviewed the effectiveness of hotlines on outbreak response in Africa”, little is known about the role and they reviewed the effectiveness. The gap isn’t answered, I would suggest authors say we systematically reviewed the role and effectiveness of hotlines in outbreak response in Africa.

2. Similarly, the title can also be highlighted as Role and effectiveness of telephone hotlines in outbreak response in Africa: a systematic review and meta-analysis

Methods

3. The authors say “during the last 25 years”, It would be good to state the actual period for example from 199x to 202x

4. “The search was also repeated on 16 September 202” please write the year correctly

Results

5. Could the authors also highlight the different outbreaks in which these telephone hotlines were used.

Introduction

6. “Public and private entities have used telephone hotlines to provide general health information and education and have also, in some cases, repurposed them as alert lines for outbreak response.” Please reference

7. “Most reviews of mobile health in low and middle income countries (LMICs) have focused on the use of mobile phones in chronic disease management and health behavior changes in non-outbreak settings (4)” the authors say most reviews but only one citation

8. “This systematic review and meta-analysis assessed the published evidence on the use of telephone hotlines for outbreaks in Africa” the authors need to be consistent; role/ use, or role and effectiveness

Methods

9. There was no restriction on language, were there any papers not in English that were included?

10. Please state the period considered for inclusion of the studies. For example, studies done between this period(xx-yy) were eligible

11. “Two independent individuals (NF and CA) searched citations in MEDLINE, Embase, PsycINFO, and Web of Science.” In the abstract, the authors state that five databases were searched including Global Health, please clarify.

Results

12. Flow chart

1251 records were screened, 1211 were excluded and 41 were assessed for eligibility, this is not adding up

13. “Effect of the hotlines” the authors ought to write this well. Effectiveness of ..

6. PLOS authors have the option to publish the peer review history of their article (what does this mean?). If published, this will include your full peer review and any attached files.

Reviewer #1: No

---

## [Author Response · Author response to Decision Letter 0]

6 Sep 2023

Queries the abstract:

Background

Query 1: The authors state “In Africa, little is known about the role of telephone hotlines in outbreak response. We systematically reviewed the effectiveness of hotlines on outbreak response in Africa”, little is known about the role and they reviewed the effectiveness. The gap isn’t answered, I would suggest authors say we systematically reviewed the role and effectiveness of hotlines in outbreak response in Africa. 

Response: The authors would like to thank the review for this comment. This has been modified in the abstract and now reads as follows:

We systematically reviewed the role and effectiveness of hotlines on outbreak response in Africa.

Query 2: Similarly, the title can also be highlighted as Role and effectiveness of telephone hotlines in outbreak response in Africa: a systematic review and meta-analysis.

Response: The title has been modified and now reads:

Role and effectiveness of telephone hotlines in outbreak response in Africa: a systematic review and meta-analysis

Methods

Query 3: The authors say “during the last 25 years”, It would be good to state the actual period for example from 199x to 202x

Response: This has been re-written as follows:

studies on the use of telephone hotlines in outbreak response in Africa published between January 1995 and August 2020.

Query 4: “The search was also repeated on 16 September 202” please write the year correctly.

Response: This has been corrected and now reads:

The search was also repeated on 16 September 2022

Results:

Query 5: Could the authors also highlight the different outbreaks in which these telephone hotlines were used.

Response: These outbreaks have been added to the results section of the abstract as follows:

The hotlines were set up for unusual respiratory events (one study), polio (one study), Ebola (10 studies), COVID-19 (two studies), malaria (one study), influenza-like illnesses (ILI) (one study) and rift valley fever in livestock (one study).

Queries on the introduction

Query 6: “Public and private entities have used telephone hotlines to provide general health information and education and have also, in some cases, repurposed them as alert lines for outbreak response.” Please reference

Response: The reference has been added. A statement using Malawi as an example was also added.

This now reads as follows:

Public and private entities have used telephone hotlines to provide general health information and education and repurposed them as alert lines for outbreak response. For example, in Malawi, the Health Centre by Phone which was used to provide health information on nutrition and general health, was later repurposed for outbreak response during the COVID-19 pandemic (4,5).

Query 7: “Most reviews of mobile health in low- and middle-income countries (LMICs) have focused on the use of mobile phones in chronic disease management and health behavior changes in non-outbreak settings (4)” the authors say most reviews but only one citation.

Response: Only one reference was given because it is a systematic review of systematic reviews. This systematic review reference here summarised the evidence from 23 systematic reviews. One additional reference has also been added. 

Query 8: “This systematic review and meta-analysis assessed the published evidence on the use of telephone hotlines for outbreaks in Africa” the authors need to be consistent; role/ use, or role and effectiveness

Response: This has been adjusted and the sentence now reads:

This systematic review and meta-analysis assessed the published evidence on the role and effectiveness of telephone hotlines for outbreaks in Africa.

Queries on the methods

Query 9: There was no restriction on language, were there any papers not in English that were included?

Response: We did not find any published study in another language such as French. However, we considered it relevant not to limit the search based on language.

Query 10: Please state the period considered for inclusion of the studies. For example, studies done between this period(xx-yy) were eligible

Response: A sentence has been added as follows:

Studies were included if they: 1) were published between January 1995 and September 2022 where the last search was carried out………

Query 11: “Two independent individuals (NF and CA) searched citations in MEDLINE, Embase, PsycINFO, and Web of Science.” In the abstract, the authors state that five databases were searched including Global Health, please clarify.

Response: Five conventional databases were searched. In the main text, we omitted Global Health. This has now been edited. It now reads:

Two independent individuals (NF and CA) searched citations in MEDLINE, Embase, PsycINFO, Global Health and Web of Science. The search strategy is in supplemental table 1. 

Query on results

Query 12: Flow chart 1251 records were screened, 1211 were excluded and 41 were assessed for eligibility, this is not adding up.

Response: There was a typo. Out of 1251 records screened, 1210 were excluded and 41 were assessed for eligibility. This has now been corrected on the flow chat.

Query 13: “Effect of the hotlines” the authors ought to write this well. Effectiveness of ..

Response: This has been corrected from effect to effectiveness.

---

## [Editor Report · Decision Letter 1]

12 Sep 2023

Role and effectiveness of telephone hotlines in outbreak response in Africa: a systematic review and meta-analysis

PONE-D-23-02720R1

Dear Dr. Fongwen,

We’re pleased to inform you that your manuscript has been judged scientifically suitable for publication and will be formally accepted for publication once it meets all outstanding technical requirements.

Kind regards,

Felix Bongomin, MB ChB, MSc, MMed, FECMM

Academic Editor

PLOS ONE
---

## [Editor Report · Acceptance letter]

15 Sep 2023

PONE-D-23-02720R1 

Role and effectiveness of telephone hotlines in outbreak response in Africa: a systematic review and meta-analysis 

Dear Dr. Fongwen:

I'm pleased to inform you that your manuscript has been deemed suitable for publication in PLOS ONE. Congratulations! Your manuscript is now with our production department. 

Kind regards, 

on behalf of

Dr. Felix Bongomin 

Academic Editor

PLOS ONE